# Impact of an Integrative Nutrition Package through Home Visit on Maternal and Children Outcome: Finding from Locus Stunting in Yogyakarta, Indonesia

**DOI:** 10.3390/nu14163448

**Published:** 2022-08-22

**Authors:** Tri Siswati, Slamet Iskandar, Nova Pramestuti, Jarohman Raharjo, Agus Kharmayana Rubaya, Bayu Satria Wiratama

**Affiliations:** 1Department of Nutrition, Politeknik Kesehatan Kemenkes Yogyakarta, Tata Bumi No. 3, Banyuraden, Gamping, Sleman, Yogyakarta 55293, Indonesia; 2Pusat Unggulan Iptek Inovasi Teknologi Terapan Kesehatan Masyarakat, Politeknik Kesehatan Kemenkes Yogyakarta, Tata Bumi No. 3, Banyuraden, Gamping, Sleman, Yogyakarta 55293, Indonesia; 3Balai Litbang Kesehatan Banjarnegara, Selamanik No. 16 A, Banjarnegara 53415, Indonesia; 4Department of Environmental Health, Politeknik Kesehatan Kemenkes Yogyakarta, Tata Bumi No. 3, Banyuraden, Gamping, Sleman, Yogyakarta 55293, Indonesia; 5Department of Epidemiology, Biostatistics and Population Health, Faculty of Medicine, Public Health and Nursing, Universitas Gadjah Mada, Yogyakarta 55281, Indonesia; 6Graduate Institute of Injury Prevention and Control, College of Public Health, Taipei Medical University, Taipei 11031, Taiwan

**Keywords:** integrative nutrition package, intervention, growth, development, children, stunting

## Abstract

**Background:** Stunting has been a public health problem in several developing countries including Indonesia. One of the strategies to reduce stunting was family assistance. This study was aimed to estimate the effect of family assistance by using an integrative nutrition package through home visits on the growth and development of stunted children. **Method:** This was an experimental study using pre-test post-test with control group design, conducted in Yogyakarta, Indonesia, on March to May 2022. The intervention group was provided an integrative nutrition package (INP) including maternal education, behavioral change through home visit, as well as monitoring children’s outcome, while the control group was asked to read and follow child care procedure in the maternal and child health (MCH) book as a standard procedure. Both groups were visited by trained health volunteers and had a complementary feeding (CF) package weekly for four weeks. The outcomes of this study were the maternal outcome (knowledge and behavior on children’s growth monitoring (CGM), children’s development monitoring (CDM), and infant/young children feeding (IYCF) as well as children’s outcomes, including body weight (BW), body height (BH), and child score development (CSD). This study used generalized estimating equation (GEE) to estimate the differences in differences (DID) of the impact of intervention compared with control group and compared among three different times (baseline, fourth, and eighth week). **Results:** There were 60 stunted children under five years in this study, i.e., 30 in intervention group and 30 in control group. From the GEE analysis, it was found that the regression adjusted DID showed statistically significant increase of all outcomes including children’s development score (CDS). The adjusted DID effect (95% CI) on 8th week for children’s weight, height, and development score were 0.31 (0.25–0.37), 0.41 (0.13–0.68), and −0.40 (−0.59–(−0.21)), respectively, among the intervention group. **Conclusions:** INP through home visit successfully increased maternal and children’s outcomes compared witsh standard procedure. The effect of intervention was found to be consistently significant in the fourth and eighth weeks after intervention. We recommend the local government to apply INP through home visit especially in high-prevalence stunting areas.

## 1. Introduction

In Indonesia, stunting remains a major nutritional problem due to its wider impact in a whole life span. According to the National Medium-Term Development Plan, we set a target of achieving a stunting prevalence of 14% by 2024 [1]. Based on the nutritional status survey of Indonesia in 2021, it was reported that the prevalence of stunted children in Yogyakarta province was 17.3% [2]. Although the stunting problem in Yogyakarta is a minor health problem [3], the issue of stunting disparity is an important issue [4]. The highest stunting prevalence in Yogyakarta is 26%, placed in the Dlingo sub-district of Bantul Regency [5].

Bantul is an area with characteristics of low economic growth below the provincial average [6], an average length of schooling of 9.55 years or equivalent to graduating from junior high school [7], and a high incidence of early marriage [8]. The majority of people in the district works in informal sector as farmers, farm laborer, and other informal occupations. Meanwhile, most of the mothers become housewives [6]. In general, socio-economic factors are strongly related with stunting children. It can be explained that the low economic level of the family is related to the limited selection of food, school facilities, and infrastructure as well as fewer opportunities to have good health care, selection of an environment that supports the growth and development of children, and the presence of health inequity. Stunted children who grow up within families of a poor socioeconomic level will give birth to intergenerational stunting. It is likely that mothers who are stunted and grow up in an unfavorable economic environment will have less knowledge, awareness, and attention to the growth and development of their children. This condition affects parenting, the search for health services, and the stimulation of their toddler’s growth and development. Subsequently, it impacts the child’s neurodevelopmental growth, health, and well-being.

In order for toddlers to reach their full growth and development potential, intervention measures are necessary. According to a number of studies, mentorship is the most effective strategy for preventing stunting in toddlers. Moreover, integrated intervention support in the form of education, communication of dietary behavior modification, supplementary feeding, and stimulation of toddlers’ growth and development is preferred above a single intervention [9]. In Presidential Regulation No. 72 of 2021 of the Republic of Indonesia, stunting reduction acceleration was issued, and it is stated that the national action plan to encourage stunting decrease is to give a support families at risk of stunting; the goal is to enhance access to information and services [10].

In this study, we conducted an intervention nutrition package (INP) consisting of maternal education and behavior change communication according to growth and development monitoring for children as well as infant/young children feeding (IYCF) and complementary feeding through home visit. It is different from the routine program using classical monitoring in Posyandu as a maternal and children’s health service in Indonesia. Thus, we wanted to investigate how an INP delivered through home visits affects maternal knowledge and behavior on children’s growth monitoring (CGM), children’s development monitoring (CDM), infant/young children feeding (IYCF), and stunted children’s body weight (BW), body height (BH), and children’s development score (CDS).

## 2. Materials and Methods

### 2.1. Study Design

This was an experimental study with a pre-test post-test with a control group design in the highest stunting prevalence area in Yogyakarta, Indonesia. We selected two villages that represented the population with the highest prevalence (Desa Muntuk and Jatimulyo) in March–May 2022.

### 2.2. Study Population and Sampling

The population comprised stunted toddlers in the two villages. By following Lemeshow’s formula for the intervention of two groups, using a significance level at 5% and assumed 80% power, a design effect of 1.5, and loss to follow-up of 10%, the number of samples per group was calculated as 30. The sampling flow chart can be seen in Figure 1.

### 2.3. Eligibility Criteria

The sample criteria were determined by selecting toddlers who were still living with their mothers, aged 6–59 months, based on complicated factors such as household income, parental education level, history of low birth weight, and close birth spacing as well as the degree of severe stunting as measured by the z-score. The researchers, dietitians, and midwives performed this assessment using a checklist. Contrarily, the exclusion criteria were: a non-poor family, a twin, and concurrent obesity and stunting.

### 2.4. Intervention Package

INP as the intervention was a set of interventions for maternal education and behavior modification. The package consists of CGM, CDM, and IYCF literacy and counseling based on children’s health problem, while subjects in the control group were only reminded to read MCH books. Both the treatment and control groups were visited once a week for four weeks and were given a CF package consisting of carbohydrate-source foods, animal diet, protein, cereals, vegetables, and fruits. In the intervention group, the home visits were carried out by health cadres who were trained in advance in the procedure. They taught and communicated the behavior change to mothers about monitoring the growth and development of toddlers, stimulating growth and development, reading growth charts, and providing appropriate food; provided counseling to mothers according to toddler health problems; and gave CF. Meanwhile, in the control group, the health visit was only to remind mothers to read MCH books and to supply CF. The cadres who made home visits were from the local Posyandu (English: integrated health service post), and lived in the same sub-village as the accompanying toddlers. The ratio of cadres to toddlers was 1:2. This intervention is different from the existing program in that the education is carried out classically at the local Posyandu monthly.

### 2.5. Study Outcomes

This study’s results were separated into two categories: direct outcomes, which included mothers’ knowledge and abilities about CGM, CDM, and IYCF, and follow-up outcomes, which included BW, BH, and CDS. All outcomes were evaluated at baseline, follow-up 1 (4 weeks later), and follow-up 2 (8th week).

### 2.6. Data Collecting and Instruments

At the baseline, we provided a questionnaire to collect social, economic, and demographic information, such as the age, education, and occupation of the parents. We assessed mother understanding and behavior regarding CGM, CDM, and IYCF. A structured questionnaire of ten questions was used to examine the mother’s knowledge, while a checklist of ten items was used to evaluate the mother’s CGM, CDM, and IYCF practices. The true score was one, and the false score was zero, for a total of 30. The impact of the intervention on child outcomes was measured in terms of BW with a baby scale, BH with a stature meter (infantometer for 2 years old and portable microtoise stadiometer for two years), and CSD with an early detection of growth and development form consisting of components of gross and fine motoric development, language, social skills, and children’s independence. The developmental achievement of toddlers is given a value of 0 if it is normal, 1 if it is questionable, and 2 if it is abnormal. In this study, cadres were responsible for data collection.

### 2.7. Data Analysis

Initially, we performed a univariate analysis to summarize the characteristics of participants in both treatment groups. Afterwards, we provided an illustrated repeated survey of maternal and children’s outcomes at baseline and follow-ups 1 and 2. Then, we examined the intervention’s influence on the outcomes, compared them over time and across groups, and made modifications. The impact of the repeated intervention was measured by utilizing the GEE model in StataCorp. 2017. Stata Statistical Software: Release 15. College Station, TX, USA: StataCorp LLC [11,12]. The reference category in this study was the control group. We estimated the difference in differences (DID), which was defined as the interaction between intervention and control difference with differences among baseline and first and second follow-ups. This study reported adjusted beta coefficients, *p*-values, and 95% confidence intervals (CI).

### 2.8. Ethics and Informed Consent

The Institutional Review Board of Politeknik Kesehatan Kemenkes, Yogyakarta, granted ethical approval for this study, No. e-KEPK/POLKESYO/0223/II/2022, dated 23 February 2022. The informed consent was signed by the toddlers’ mothers.

## 3. Results

### 3.1. Characteristics of Participants

Maternal age, father’s and mother’s education, and father’s and mother’s work were similar in the intervention and control groups although the control group included more low-income households. The characteristics of children under five, including age and gender, were comparable across the two intervention groups. Baseline measurements of maternal knowledge and practice on CGM, CDM, and IYCF and BW, BH, and CDS did not reveal any significant differences between the two groups. Specific details are provided in Table 1.

### 3.2. The Effect of Intervention on Maternal and Children’s Outcomes among Different Times

The intervention had a positive effect on all maternal outcomes in general, with the treatment group demonstrating a greater gain in knowledge scores of CGM, DGM, and IYCF than the control group (Figure 2, Figure 3 and Figure 4). The outcomes of this study were also observed in maternal practice, with larger increases in CGM, DGM, and IYCF scores in the treatment group than in the control group (Figure 5, Figure 6 and Figure 7). This finding is comparable to the impact of the intervention on children under five, including BW, BH, and CSD; however, the improvement of the effect in the treatment group was greater than that in the control group (Figure 8, Figure 9 and Figure 10).

### 3.3. Association between Intervention and Research Outcome among Different Times

The analysis in Table 2 reveals the association between intervention and research outcomes among different observational times. The results of the study show that there were no significant differences between treatments on baseline measurements, while in the first follow-up measurements, there were significant differences in maternal outcomes about the aspect of toddler growth knowledge. On the other hand, in the second follow-up, there were significant differences in maternal outcomes in the aspect of maternal knowledge about children growth and development as well as of children’s outcomes in the aspect of children’s development.

### 3.4. Association between Time and Research Outcome among Different Intervention Groups

The analysis, in Table 3, reveals that the INP had a significant influence on maternal outcomes, including CGM, CDM, and IYCF, at follow-ups 1 and 2 and that mother practice related to CGM, as measured by the overall practice score, was superior at follow-ups 1 and 2 compared to before the intervention. The control group demonstrated significant changes at follow-up 1 only in total knowledge, CDM, and total practice scores but at follow-up 2 only in total practice relative to before the intervention. In general, during both follow-ups 1 and 2, all children’s outcomes for each group intervention exhibited substantial favorable effects.

### 3.5. Evaluation of the Intervention on Research Outcomes Based on General Estimating Equation Analysis

In Table 4 below, after full adjustment, the analysis showed a significant interaction between intervention and time for all research outcomes by group intervention. The INP through home visit had a consistently significant positive effect after follow-ups 1 and 2 on all research outcomes, especially children’s body weight, height, and developmental score. The INP through home visit group had higher weight and height gain on follow-up 2 compared with the control group. The INP through home visit group also had a better developmental score change on follow-up 2 compared with the control group. The detailed results are shown in Table 4.

## 4. Discussion

According to the findings of this study, the two groups shared all of the observed features, confirming that the impact was a result of the intervention administered. The impact of the INP’s 4-week and 8-week interventions on maternal knowledge and practice in CGM, CDM, and IYCF through home visits is encouraging. Moreover, the children’s outcome indicator as a result of the INP four and eight weeks later in both the intervention and control groups revealed a beneficial effect. However, the intervention group showed better improvement in all evaluated outcomes than the control group. After performing the DID analysis and adjusting for numerous factors by comparing the intervention group to the control group, this conclusion revealed a stable improvement in all intervention group outcomes.

It can be explained that the intervention group had greater maternal and children’s outcome outcomes than the control group because (1) home visits give chances for more extensive interaction between cadres and mothers of toddlers [13] and can bridge the gap between ignorance and mistakes in the practice of maternal parenting on the health of toddlers [14,15]; (2) the intervention-supplied knowledge and practice from cadres to mothers is a positive mediator [9,16]; (3) the length of each home visit in this study was 45–60 min longer than the control (15–30 min); (4) the uniformity of home visits was assured through supervision; and (5) IYCF teaching paired with food ingredients enabled parents to instantly implement proper feeding practices for their children [16]. At follow-ups 1 and 2, the INP intervention significantly increased BW compared to the control group. This is related to the effect size of the intervention on maternal outcomes compared to the control group. It was stated in previous studies that maternal knowledge and dietary practices in accordance with recommendations are closely related to children’s health [17].

This result is similar with the research conducted by Paramasanti and Sulistyawati in Yogyakarta, Indonesia [18]; Saleemi et al. in peri-urban Pakistan [19]; and Zaman et al. in Lahore, Pakistan [20]. In contrast to the control group, the impact of INP on BH children was substantial at follow-up 2. A prospective investigation in Uganda revealed there was no correlation between CF and HAZ, and this study highlighted the necessity for longer surveillance [21]. As mentioned in earlier research [20,21,22], CF dietary variety can encourage linear development in stunted children, as shown in this study.

Similarly proven by previous research in India [23], diet is a crucial element in the development of toddlers [24,25]. Cadres presented mothers with instances of developmental stimulation during home visits, and mothers implemented it directly under the supervision of cadres. The impact of INP was statistically significant and bigger than that of the control group because, at that time, toddlers’ mothers assessed their babies’ developmental scores while continually stimulating their developmental progress until follow-ups 1 and 2. Previous research in India [23] and Peru [11] indicated that toddlers’ developmental scores were enhanced by developmental stimulation. Moreover, the intervention group received CF in conjunction with developmental stimulation, allowing for a greater probability of improving CDS than the control group, which received simply CF and a reminder to read MCH books [25].

This study possesses several strengths. First, educational intervention by cadres raised mothers’ understanding and behavior change communication through home visits on CGM, CDM, and IYCF and also allowed determining their impacts on both maternal and children’s outcomes (BW, BH, SDC). In order to characterize the change in effects following the intervention and follow-up, weekly repeated measures were performed. The second benefit is that the integrative package allows for numerous impacts rather than a single intervention. Thirdly, this strategy also includes CF so that mothers can directly practice correct feeding. According to a 2007 Lancet study, an integrated package consisting of nutritional intervention, developmental stimulation, and health is highly suggested since it offers more advantages to counteract the impacts of early malnutrition [11]. Fourth, this intervention is offered to infants and toddlers in order to maximize the potential leverage to overcome the impacts of stunting in later life [12,26]. Early infancy is a crucial time for neuroplasticity to begin, which is connected to the brain’s responsiveness to stimuli [27]. Moreover, brain ability is greatly impacted by several factors, such as availability of food, nutritional state, and relationship between mother and child [13,14,18,27], which this intervention provides.

However, our study has a number of limitations, including the fact that, despite our 8-week follow-up, the evaluation of long-term monitoring after 8 weeks is crucial. Both of these studies are community-based; therefore, the effectiveness of stunting prevention is highly dependent on the knowledge and consistency of the subjects, both the cadres and the mothers, in providing proper child care. The three IYCF practices are largely reliant on food availability, food access, and purchasing power. In this study, we supplied CF so women were able to practice proper child feeding. The practical and policy implications of this study’s findings enable the continuation of INP through home visits through multi-stakeholder and multi-sectoral collaboration. The last limitation on the generalizability of this study is that all participants are Javanese.

## 5. Conclusions

With the exception of CH, the integrated nutrition package and behavior treatments in this study of stunted children improved mothers’ knowledge and practice of CGM, CDM, and IYCF as well as children’s outcomes related to BW and mending SDC at follow-up periods 1 and 2. However, additional studies are required to analyze the performance of intervention nutrition packages and different combinations of interventions to determine which are the most successful and cost-effective. Such future study would aid in the creation and development of the most effective and efficient methods for achieving Indonesia’s current nutrition goals for children.

## Figures and Tables

**Figure 1 nutrients-14-03448-f001:**
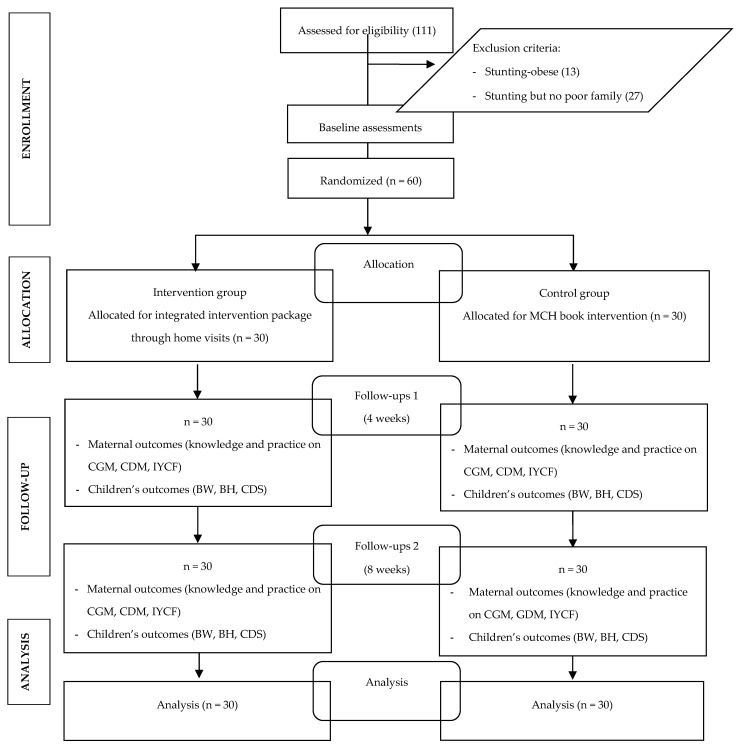
CONSORT diagram of this study.

**Figure 2 nutrients-14-03448-f002:**
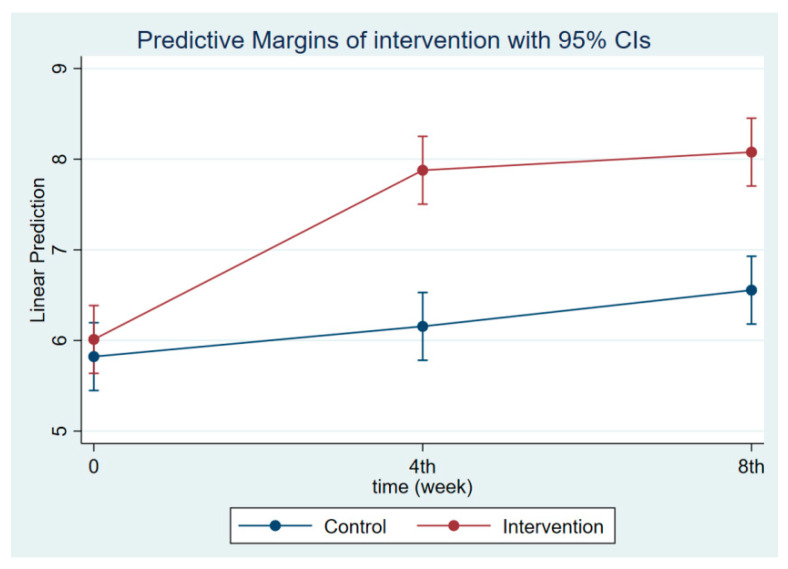
Maternal knowledge regarding CGM at follow-ups 1 and 2.

**Figure 3 nutrients-14-03448-f003:**
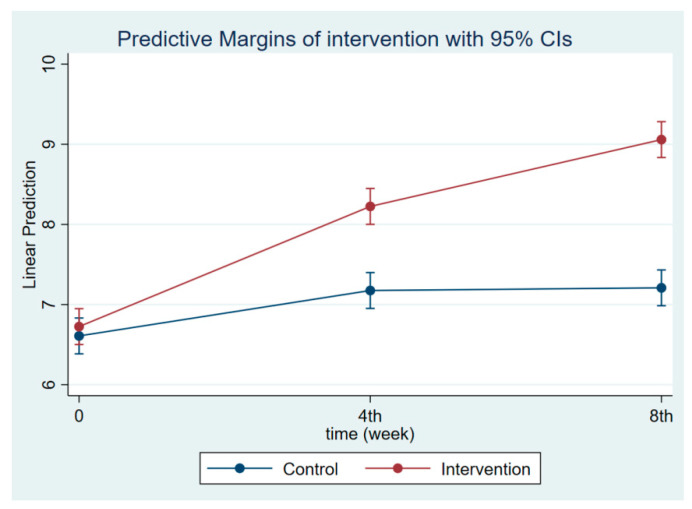
Maternal knowledge regarding CDM at follow-ups 1 and 2.

**Figure 4 nutrients-14-03448-f004:**
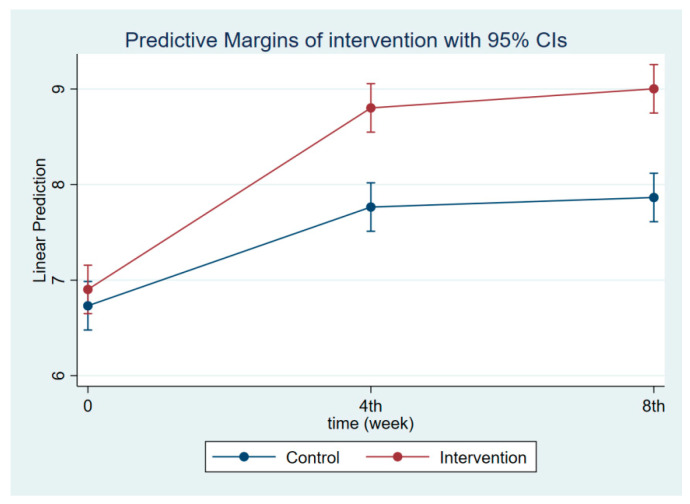
Maternal knowledge regarding IYCF at follow-ups 1 and 2.

**Figure 5 nutrients-14-03448-f005:**
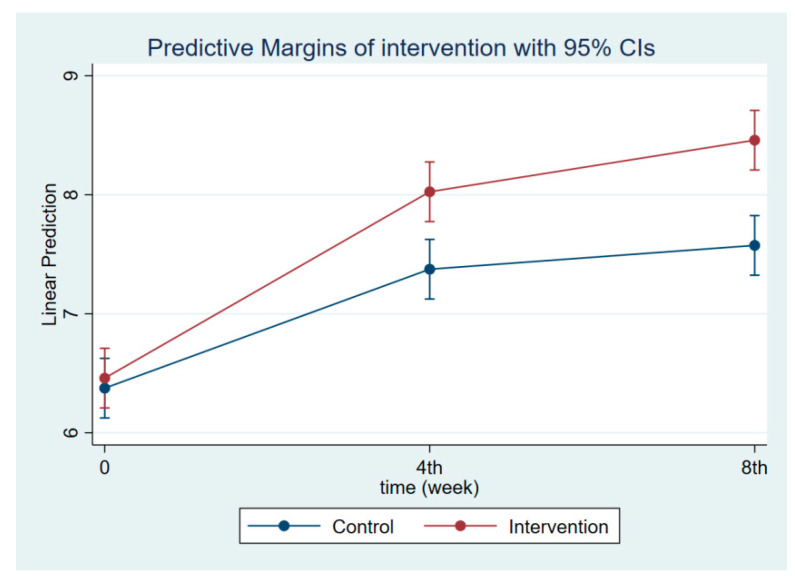
Maternal practice regarding CGM at follow-ups 1 and 2.

**Figure 6 nutrients-14-03448-f006:**
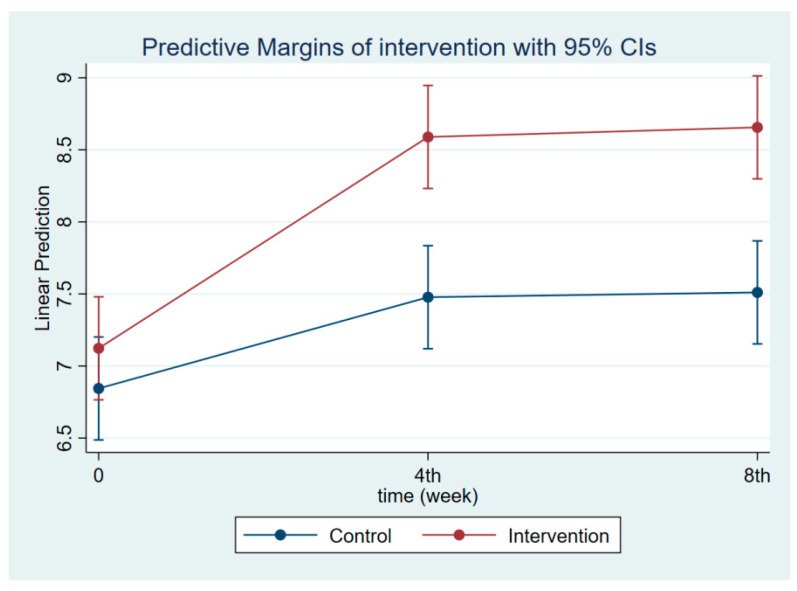
Maternal practice regarding CDM at follow-ups 1 and 2.

**Figure 7 nutrients-14-03448-f007:**
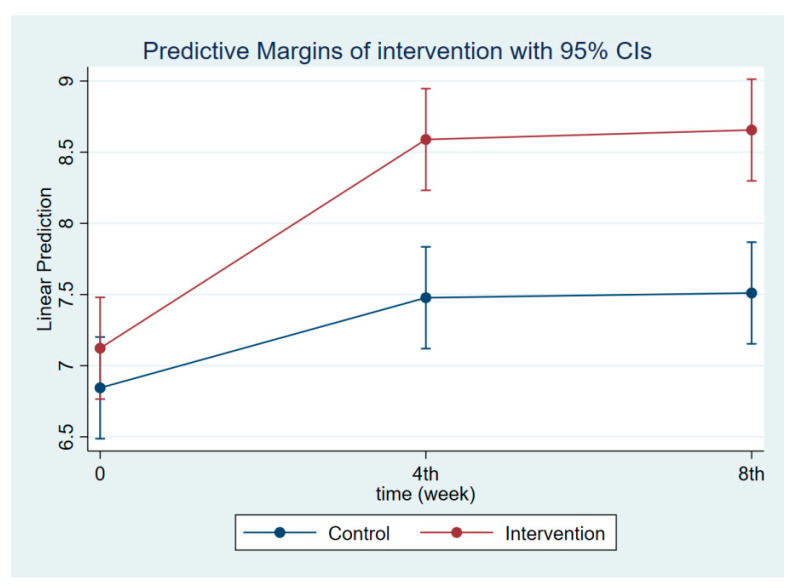
Maternal practice regarding IYCF at follow-ups 1 and 2.

**Figure 8 nutrients-14-03448-f008:**
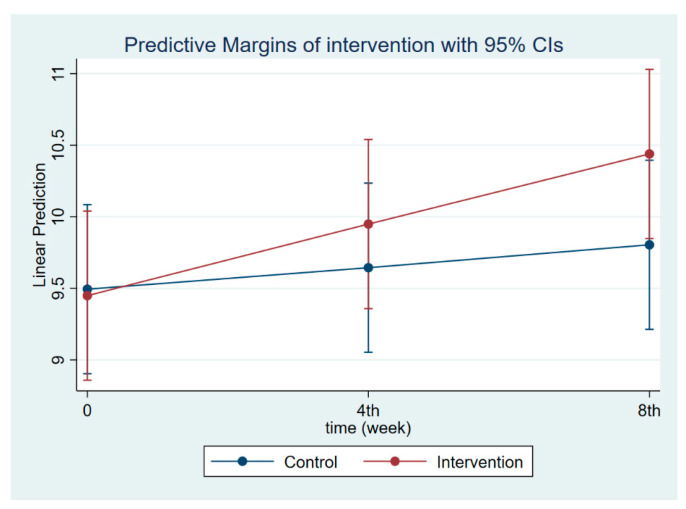
Children’s body weight at follow-ups 1 and 2.

**Figure 9 nutrients-14-03448-f009:**
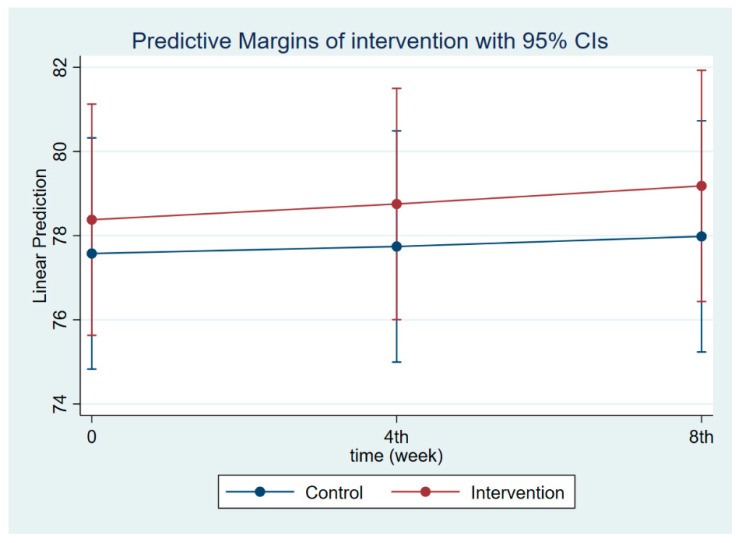
Children’s body height at follow-ups 1 and 2.

**Figure 10 nutrients-14-03448-f010:**
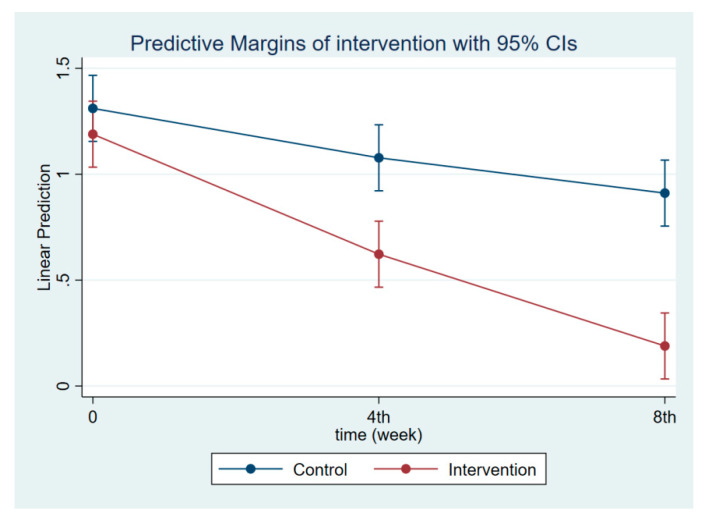
Children’s development scores at follow-ups 1 and 2.

**Table 1 nutrients-14-03448-t001:** Maternal and children characteristics by treatment group.

Variables	Intervention (n = 30)	Control (n = 30)	*p*-Value ^1^
Mother’s age			0.76
<20 years	1 (3.3%)	2 (6.7%)	
20–30 years	17 (56.7%)	11 (36.6%)	
>31 years	12 (40.0%)	17 (56.7%)	
Mother’s educational level			0.78
Elementary school (finished)	2 (6.7%)	2 (6.7%)	
Junior high school (finished)	10 (33.3%)	13 (43.3%)	
Senior high school (finished)	17 (56.7%)	13 (43.3%)	
Diploma/university (finished)	1 (3.3%)	2 (6.7%)	
Father’s educational level			0.41
Elementary school (finished)	5 (16.7%)	4 (13.3%)	
Junior high school (finished)	9 (30.0%)	11 (36.7%)	
Senior high school (finished)	16 (53.3%)	13 (43.3%)	
Diploma/university (finished)	0 (0%)	2 (6.7%)	
Mother’s occupation			0.24
Farmer and farm worker	2 (3.3%)	1 (3.3%)	
Private employee, entrepreneur	2 (6.7%)	1 (3.3%)	
Housewife	24 (80.3%)	28 (93.4%)	
Others	2 (6.7%)	0 (0%)	
Father’s occupation			0.74
Farmer and farm worker	17 (56.3%)	17 (56.3%)	
Private employee, entrepreneur	5 (16.7%)	5 (16.7%)	
Civil servant	5 (16.7%)	2 (6.7%)	
Others	3 (10.3%)	6 (20.0%)	
Economic status			0.19
Poor family	11 (36.7%)	16 (53.3%)	
Non-poor family	19 (63.3%)	14 (46.7%)	
Child’s age (moths)			1.00
≤24 months	18 (60.0%)	18 (60.0%)	
>24 months	12 (40.0%)	12 (40.0%)	
Child’s gender			0.592
Male	12 (40.0%)	10 (33.3%)	
Female	18 (60.0%)	20 (66.7%)	
Maternal knowledge			
CGM	6.03 ± 1.25	5.80 ± 1.40	0.49
CDM	6.73 ± 0.83	6.60 ± 0.72	0.51
IYCF	6.93 ± 0.64	6.70 ± 1.09	0.32
Mean score	6.57 ± 0.60	6.37 ± 0.59	0.20
Maternal practice			
CGM	6.47 ± 1.04	6.37 ± 0.67	0.66
CDM	6.77 ± 0.94	6.67 ± 0.76	0.65
IYCF	7.10 ± 1.54	6.87 ± 0.97	0.49
Mean score	6.78 ± 0.64	6.63 ± 0.55	0.35
Children Outcome			
BW (kg)	9.47 ± 2.09	9.47 ± 1.68	0.99
BH (cm)	77.86 ± 9.93	78.09 ± 8.37	0.92
CDS	1.20 ± 0.48	1.30 ± 0.59	0.48

^1^ Chi-square or fisher exact test for categorical data; independent *t*-test for numeric data. *p* < 0.05 indicates statistical significance.

**Table 2 nutrients-14-03448-t002:** Association between intervention and research outcome among different times.

Variables	Intervention	Control	Mean Difference (95% CI)
Baseline			
Maternal knowledge regarding			
CGM	6.03 ± 1.25	5.80 ± 1.40	0.23 (−0.45–0.92)
CDM	6.73 ± 0.83	6.60 ± 0.72	0.13 (−0.27–0.54)
IYCF	6.93 ± 0.64	6.70 ± 1.09	0.23 (−0.23–0.69)
Maternal practice regarding			
CGM	6.47 ± 1.04	6.37 ± 0.67	0.10 (−0.35–0.55)
CDM	6.77 ± 0.94	6.67 ± 0.76	0.10 (−0.34–0.54)
IYCF	7.10 ± 1.54	6.87 ± 0.97	0.23 (−0.43–0.89)
Children’s outcome			
BW	9.47 ± 2.09	9.47 ± 1.68	−0.03 (−0.98–0.98)
BH	77.86 ± 9.93	78.09 ± 8.37	−0.23 (−4.98–4.51)
CDS	1.20 ± 0.48	1.30 ± 0.59	−0.1 (−0.38–0.18)
Follow-up 1			
Maternal knowledge regarding			
CGM	7.90 ± 0.55	6.13 ± 1.48	1.77 (1.19–2.34) *
CDM	8.23 ± 0.43	7.17 ± 0.75	1.07 (0.75–1.38)
IYCF	8.83 ± 0.70	7.73 ± 0.87	1.10 (0.69–1.51)
Maternal practice regarding			
CGM	8.03 ± 0.61	7.37 ± 0.72	0.67 (0.32–1.01)
CDM	8.03 ± 0.56	7.13 ± 0.86	0.90 (0.53–1.27)
IYCF	8.57 ± 0.73	7.50 ± 1.28	1.07 (0.53–1.60)
Children’s outcome			
BW	9.97 ± 2.12	9.62 ± 1.65	0.35 (−0.63–1.33)
BH	78.23 ± 9.95	78.26 ± 8.36	−0.03 (−4.78–4.72)
CDS	0.63 ± 0.49	1.07 ± 0.52	−0.43 (−0.69–(−0.17)) *
Follow-up 2			
Maternal knowledge regarding			
CGM	8.10 ± 0.31	6.53 ± 0.97	1.57 (1.19–1.94) *
CDM	9.07 ± 0.58	7.20 ± 0.61	1.87 (1.55–2.18) *
IYCF	9.03 ± 0.61	7.83 ± 0.53	1.20 (0.90–1.49)
Maternal practice regarding			
CGM	8.47 ± 0.63	7.57 ± 0.57	0.90 (0.59–1.21)
CDM	8.33 ± 1.12	7.43 ± 0.73	0.90 (0.41–1.39)
IYCF	8.63 ± 0.85	7.53 ± 0.68	1.10 (0.70–1.50)
Children’s outcome			
BW	10.46 ± 2.08	9.78 ± 1.65	0.68 (−0.29–1.65)
BH	78.66 ± 9.96	78.50 ± 8.35	0.16 (−4.59–4.91)
CDS	0.20 ± 0.41	0.90 ± 0.40	−0.7 (−0.91–(−0.49)) *

* *p*-value < 0.05

**Table 3 nutrients-14-03448-t003:** Association between time and research outcome among different intervention groups.

Variables	Pretest	Post-Test 1	Post-Test 2	Post-Test 1 vs. Pretest (95% CI)	Post-Test 2 vs. Pretest (95% CI)
Intervention group					
Maternal knowledge					
CGM	6.03 ± 1.25	7.90 ± 0.55	8.10 ± 0.31	1.87 (1.38–2.35) *	2.07 (1.61–2.53) *
CDM	6.73 ± 0.83	8.23 ± 0.43	9.07 ± 0.58	1.50 (1.16–1.84) *	2.33 (1.96–2.70) *
IYCF	6.93 ± 0.64	8.83 ± 0.70	9.03 ± 0.61	1.90 (1.58–2.22) *	2.10 (1.78–2.42) *
Maternal practice					
CGM	6.47 ± 1.04	8.03 ± 0.61	8.47 ± 0.63	1.57 (1.07–2.06) *	2.00 (1.49–2.51) *
CDM	6.77 ± 0.94	8.03 ± 0.56	8.33 ± 1.12	1.27 (0.85–1.68)	1.57 (0.99–2.14)
IYCF	7.10 ± 1.54	8.57 ± 0.73	8.63 ± 0.85	1.47 (0.80–2.13)	1.53 (0.99–2.08)
Children’s outcome					
CW	9.47 ± 2.09	9.97 ± 2.12	10.46 ± 2.08	0.50 (0.45–0.55) *	0.99 (0.91–0.99) *
CH	77.86 ± 9.93	78.23 ± 9.95	78.66 ± 9.96	0.37 (0.28–0.46) *	0.80 (0.59–0.92) *
CDS	1.20 ± 0.48	0.63 ± 0.49	0.20 ± 0.41	−0.58 (−0.75–(0.38)) *	−1.00 (−1.24–(−0.76)) *
Control group					
Maternal knowledge					
CGM	5.80 ± 1.40	6.13 ± 1.48	6.53 ± 0.97	0.33 (−0.32–0.99)	0.73 (0.08–1.39)
CDM	6.60 ± 0.72	7.17 ± 0.75	7.20 ± 0.61	0.57 (0.17–0.97)	0.60 (0.21–0.99) *
IYCF	6.70 ± 1.09	7.73 ± 0.87	7.83 ± 0.53	1.03 (0.47–1.60)	1.13 (0.65–1.62)
Maternal practice					
CGM	6.37 ± 0.67	7.37 ± 0.72	7.57 ± 0.57	1.00 (0.62–1.38)	1.20 (0.84–1.56)
CDM	6.67 ± 0.76	7.13 ± 0.86	7.43 ± 0.73	0.47 (0.03–0.90) *	0.77 (0.34–1.19)
IYCF	6.87 ± 0.97	7.50 ± 1.28	7.53 ± 0.68	0.63 (0.11–1.16)	0.67 (0.28–1.05)
Children’s outcome					
CW (kg)	9.47 ± 1.68	9.62 ± 1.65	9.78 ± 1.65	0.15 (0.10–0.20) *	0.31 (0.24–0.38) *
CH (cm)	78.09 ± 8.37	78.26 ± 8.36	78.50 ± 8.35	0.17 (0.14–0.19) *	0.41 (0.13–0.68) *
CDS	1.30 ± 0.59	1.07 ± 0.52	0.90 ± 0.40	−0.23 (−0.39–(0.07)) *	−0.40 (−0.61–(−0.19)) *

* *p*-value < 0.05.

**Table 4 nutrients-14-03448-t004:** Evaluation of the interaction effect between intervention and time on different research outcomes using adjusted DID based on GEE model.

Variables	Intervention vs. Control ^a^	
	Follow-Up 2 vs. Baseline	Follow-Up 1 vs. Baseline
Knowledge regarding CGM ^1^	1.33 (0.64–2.03) *	1.53 (0.92–2.15) *
Knowledge regarding CDM ^1^	1.73 (1.29–2.16) *	0.93 (0.49–1.38) *
Knowledge regarding IYCF ^1^	0.97 (0.48–1.46) *	0.87 (0.34–1.39) *
Practice regarding CGM ^1^	0.80 (0.31–1.29) *	0.57 (0.09–1.05) *
Practice regarding CDM ^1^	0.80 (0.23–1.37) *	0.80 (0.24–1.36) *
Practice regarding IYCF ^1^	0.87 (0.18–1.55) *	0.83 (0.18–1.49) *
Children’s body weight ^2^	0.68 (0.59–0.77) *	0.35 (0.29–0.41) *
Children’s body height ^2^	0.40 (0.01–0.79) *	0.21 (−0.07–0.48)
Children’s developmental score ^2^	−0.60 (−0.87–(−0.32)) *	−0.33 (−0.57-(−0.10)) *

* *p* < 0.05. ^a^ Adjusted difference in differences (DID) coefficients using GEE. ^1^ Adjusted with mother’s age, mother’s educational level, mother’s occupation, father’s educational level, father’s occupation, and family status; ^2^ adjusted with mother’s age, mother’s educational level, mother’s occupation, father’s educational level, father’s occupation, family status, child’s age, and child’s gender.

## Data Availability

All data and models of study are available from the corresponding author upon reasonable request.

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
