# Peer review of "Impact of an Integrative Nutrition Package through Home Visit on Maternal and Children Outcome: Finding from Locus Stunting in Yogyakarta, Indonesia"

_nutrients, 2022, doi:10.3390/nu14163448_

Round 1

Reviewer 1 Report

Overall, it is a good intervention study and well conducted and well written.

Introduction

1.       Better to include a paragraph the current maternal and childcare package in the field and existing home visiting component. It is difficult to understand the differences between existing package and intervention. Better to summarise the Line 59-77 and include current MCH services of Indonesia.   

Methods:

2.       It is important to add the sampling and the details about introduction of packages for each group. That part is not clear in this write up. It will be very useful for readers to duplicate this type of intervention package.   (Line 112-118) It is suggested to write clearly the interventional packages of each group. Component of CF assessed.

3.       Need to add the details of home visits to differentiate it from the control group.

4.       Line 136-137 – who are these intermediate cadres. Are they involved with the routine activities? Were they trained?

Results

5.       Table 1: Poor families are higher in control group please re check the significance test.   

Discussion

6.       Line 225-228 – not relevant to this study, need to adjust.

7.       Line 267-268 – Need to provide three element of CF in the methodology

8.       Line 268-269 – provision of CF is not mentioned in the methodology

Conclusion

9.       When the suggested changes in the methodology sections are, conclusions are tallying with the study.

Author Response

Dear Review

We have  revised our manuscript in accordance with the reviewers’ comments and suggestions. We appreciate all of comments.

Please kindly see my revise.

Thank you

Regards

Reviewer 2 Report

Review notes

Abstract

The title in lines 2-6 is too long hence making it difficult to follow. Please rephrase and make it concise.

There is a combination of present and past tenses in the abstract, please use one tense, lines 19-22 are present while the following sentences are past tense.

The end of sentence 22 should read May, not may. And the sentence needs to be rephrased to make it clearer

The grammar in the abstract needs to be rephrased to make it clearer. It reads like it was interpreted and missing some conjunctions hence harder to follow. This is more apparent in line 26-39.

Introduction

Rephrase paragraph 1, while all the literature is available, it is arranged poorly hence hard to follow.

DIY is mentioned in lines 51 and 52 but has not been defined prior

Rephrase sentence in lines 56-58, poor grammar

Sentence in lines 60-63 has 3 different wordings for poor education attainment linked to stunting, either choose one or rephrase to explain why all 3 are needed:

Line 60 poor parental education

Line 62-63 low knowledge

Line 63 inadequate health and education facilities

In sentence from lines 66-69, seeking health services is mentioned in line 67 and repeated in line 68

Line 81 mentions PMT but it is not defined prior

Materials and methods

Use one tense in lines 92-94, its has past and present tenses

Consider rephrasing line 97 and rearranging the whole paragraph to give details as per heading

Exclusion criteria not clear in the flow chart and eating is misspelled

Rephrase sentence in line 105-106

Rephrase sentence in lines 117 and 118

Check tenses/grammar in paragraph with lines 125 to 137

What does cadres of intermediate rank mean in lines 136-137? consider rephrasing to more common terms

Check tense and drama in paragraph in lines 139 to 148

Define the full names of the ethic boards in line 150.

Results

Figures 2-10 aren’t very clear, I hope better versions will be published, with better labelling. Information on the statistical differences between the control and the intervention groups need to be mentioned. Figure 10 indicates that the control group had a higher CDS score, this is not noted.

Consider adding statistics to section 3.3 in lines 178 to 185

It would have been great to see the comparison between control and intervention groups on the pretest, and post tests 1 and 2. The current comparison is focused on within group (Table 2). The results indicate that there were improvements in both groups for some outcomes, was this the same in the groups? Table 3 compare them after adjustment but does not provide p values, it only mentions that intervention was better than control. You could also compare follow up 1 and 2 to assess if longer intervention influences outcomes.

Discussion

The discussion section is good with all the relevant information, I would recommend improving grammar and repositioning the sentences to encourage flow of information.

Conclusion

Consider rephrasing to make a concise presentation of your findings and what you think the implications are for children health or stunting mitigation.

General comment

This is great work, it needs work on grammar and sentence presentation to flow.

Author Response

Dear reviewer,

We have  revised our manuscript in accordance with the reviewers’ comments and suggestions. We appreciate for all suggestion and please kindly see my revise. We hope further reviewer input and consideration.

Thank you
